# Dynamic OverCloud: Realizing Microservices-Based IoT-Cloud Service Composition over Multiple Clouds

**Jungsu Han** [1] **, Sun Park** [2] **and JongWon Kim** [2,*]

1   School of Electrical Engineering and Computer Science, Gwangju Institute of Science and Technology (GIST), Gwangju 61005, Korea; jshan@nm.gist.ac.kr
2   AI Graduate School, Gwangju Institute of Science and Technology (GIST), Gwangju 61005, Korea; sunpark@nm.gist.ac.kr
*   Correspondence: jongwon@nm.gist.ac.kr; Tel.: +82-62-715-2219

**Abstract:** With the expansion of cloud-leveraged Information and Communications Technology (ICT) convergence trend, cloud-native computing is starting to be the de-facto paradigm together with MSA(Microservices Architecture)-based service composition for agility and efficiency. Moreover, by bridging the Internet of Things (IoT) and cloud together, a variety of cloud applications are explosively emerging. As an example, the so-called IoT-Cloud services, which are cloud-leveraged inter-connected services with distributed IoT devices, dynamically utilize geographically-distributed multiple clouds since mobile IoT devices can selectively connect to the near-by cloud resources for low-latency and high-throughput connectivity. In comparison, most public cloud providers may cause vendor lock-in problems that limit the inter-operable service compositions. Thus, in this paper, we propose a new overlay approach to address the above limitations, denoted as Dynamic OverCloud, which is a specially-arranged razor-thin overlay layer that provides users with an inter-operable and visibility-supported environment for MSA-based IoT-Cloud service composition over the existing multiple clouds. Then, we design a software framework that dynamically builds the proposed concept. We also describe a detailed implementation of the software framework with workflows. Finally, we verify its feasibility by realizing a smart energy IoT-Cloud service with the suggested operation lifecycle.

**Keywords:** cloud-native; multiple clouds; microservices architecture; service composition; IoT-Cloud service; visibility; DataLake

---

## 1. Introduction

Cloud-based ICT technology, dominated by hyper-scale cloud giants such as Amazon, Microsoft, and Google, is becoming the core piece of future ICT infrastructure [1]. Also, with the growing popularity of the Internet of Things (IoT), various IoT services are rapidly increasing in the field of home, healthcare, factory, and farm. With this kind of cloud growth, a service composition for developing cloud applications is evolving toward an MSA-based service composition with the concept of cloud-native computing. A service composition refers to a combination of steps such as resource provisioning, resource placement, function deployment, and function stitching to complete an entire service [2]. Microservices are defined as small-sized functions that may be deployed and scaled independently of each other, and they may employ different middleware stacks for their implementation [3]. In particular, legacy services based on monolithic architecture are being migrated to containerized MSA, in order to adapt to technology changes and reduce time-to-market [4]. For the MSA-based service composition, adopting containerization over virtual machines (VMs) is gradually increasing, since it has benefits over traditional VMs in the cloud in terms of size and flexibility.

Most cloud vendors already provide tools to support the MSA-based service composition, which may cause vendor lock-in. The cloud vendor lock-in problem means the situation where users are too dependent on one cloud provider and cannot move around among cloud vendors without substantial costs, legal constraints, and technical incompatibilities [5]. The lock-in problem is evident in that applications developed for specific clouds cannot easily be migrated to other cloud platforms [6,7]. The heterogeneous nature of cloud APIs is not technically compatible, and it leads to interoperability and portability challenges [8]. And, diversified cloud applications are moving towards broader distribution across multi-clouds and the inclusion of various IoT devices, as evident through IoT networking integration in the context of edge computing [9]. Especially application's mobility and interoperability are critical issues in the next-generation mobile networks. With the evolution of edge clouds/mobile networks, agile and flexible service operations based on interoperability will emerge as an important requirement in the future.

For this reason, we propose a new approach called Dynamic OverCloud, which enables a specially-arranged razor-thin overlay layer extending our previous work in [10]. The specific contributions of this paper are summarized below.

- First, we define the concept of the Dynamic OverCloud approach, which is an additional layer located between the service and resource layers. Dynamic OverCloud provides a consistent overlay layer in a cloud-agnostic way. We design its components such as Interface Proxy, Dev+Ops Post, Cloud-native Clusters, Assembled DataLake, and Visibility Fabric to enable an inter-operable and visibility-supported environment for MSA-based service composition over multiple clouds. And, we also design a software framework that dynamically builds Dynamic OverCloud.
- Second, to make concrete our proposed concept, we implement the software framework by adopting a workflow scheme to specify its order of realization. The workflow scheme is a set of dependent or independent tasks represented in the form of a Directed Acyclic Graph (DAG), where the nodes represent the tasks and a directed edge denotes the dependency between the corresponding tasks to achieve the automated, flexible provisioning [11,12]. The implemented software framework facilitates automated and effective Dynamic OverCloud provisioning by describing well-defined tasks with the workflow.
- Third, to verify the feasibility of Dynamic OverCloud, we suggest an operation lifecycle with the implemented software framework to guide IoT-Cloud service composition using Dynamic OverCloud. Then, we apply a use case of smart energy IoT-Cloud service by following the operation lifecycle. We appear that the instance of Dynamic OverCloud is dynamically deployed with the software framework in an automated and efficient way. We also confirm that Dynamic OverCloud provides users with functionalities such as multi-layer visibility and persistent storage that assist inter-operable service composition.

The rest of this paper is organized as follows. In Section 2, we briefly summarized related work. We then design the concept of Dynamic OverCloud in Section 3. Section 4 discusses the implementation of the software framework with the workflows. In Section 5, we verify smart energy IoT-Cloud service by following the operation lifecycle over multiple clouds. Finally, we conclude this paper in Section 6.

## 2. Related Work

The cloud vendor lock-in is a challenging issue that requires substantial efforts to overcome the existing barriers for operating and developing cloud applications [13–15]. Many kinds of researches have been adopted to solve the vendor lock-in issue for cloud interoperability. The resource-oriented federation option, called as cloud service broker (CSB) can help developers to select the most appropriate cloud provider(s) in terms of functionality and quality of service (QoS) requirements [16]. Karim et al. [17] proposed a vendor-independent cloud provisioning tool that utilizes ontology-based semantic reasoning to acquire the best available cloud resources for cloud users. Kyriakos Kritikos et al. [18] proposed a cross-level and multi-cloud application adaption architecture,

which allows us to specify advanced adaption rules and histories for multi-cloud applications. In [19], CloudMF relies on a model-driven approach. It has the principle of "model once, generate anywhere" for the management of multi-cloud applications. In [20], Merle et al. propose Open Cloud Computing Interface (OCCI), open standards for managing any kinds of cloud resources. OCCI is a RESTful API for all types of cloud management tasks. It acts as a service front-end to a cloud provider's internal management framework. Sandobalin et al. [21] propose ARGON, which is an infrastructure modeling tool for cloud provisioning that leveraged Model-Driven Engineering to provide a uniform, cohesive, and seamless process with which to support the DevOps approach. OpenStack TripleO is open-source software for OpenStack deployment & management tools. It uses terms of Overcloud and Undercloud to build cloud (Overcloud) for any purpose given by the prepared OpenStack cloud (Undercloud) [22].

Meanwhile, researches on lightweight virtualization have been conducted by considering the characteristics of cloud applications for the MSA-oriented service composition. Nane Kratzke proposed a concept called lightweight virtualization cluster (LVC) relying on operating system virtualization to overcome subliminal generated vendor lock-in [23]. In similarly, Hadley et al. proposed a MultiBox framework for vendor-independent multi-cloud deployments. The suggested framework allows its users to deploy and migrate almost any application in its normal state with minimal computational and network resources overheads [24]. It is noticeable to overcome vendor lock-in of multi-clouds deployment using container technologies. In [25], the authors propose the CloudLaunch for discovering and launching cloud applications on multiple cloud providers. It allows MSA-based service composition with each application having its customizable user interface and control over the launch process while preserving cloud-agnosticism so that authors can easily make their applications available on multiple clouds with minimal effort. A commercial software, Terraform, Consul, Vault, and Nomad provided by HashiCorp, serves automated resource provisioning in terms of infrastructure as a code over multiple clouds as well as supports the service composition [26].

However, most of the above works are lacking in an attempt to systematically resolve both vendor lock-in and the dynamic service composition issues. The efforts to systematically address multiple issues from a comprehensive point of view are insufficient in researches since researches mostly focus on addressing specific issues individually, such as cloud interoperability and the MSA-based service composition. For example, in the case of the work [16], it still supports only the purpose of helping to choose cloud resources by focusing on cost-efficient algorithms. In [17–21], researches tackle the issues for cloud interoperability, but they do not directly address the MSA-based service composition. In [22], it only focuses on the management of limited OpenStack-based clouds, instead of focusing on multi-clouds management for cloud interoperability. In [23,24], though they tackle vendor lock-in issues by leveraging the containerization concept, they do not go deep to MSA-based service composition. In [25], they deal with interoperability with multiple clouds in a similar direction to our research. They leverage not only virtual appliance but also container appliance. However, they do not directly tackle service composition with the MSA concept. In [26], it provides a full-package range from IaaS to PaaS in cloud computing. This solution is suitable for big enterprises who want to solve everything at once. However, in terms of IT-operation capabilities, it can be a costly burden for developers of small services who quickly develop and realize the service based on a microservices architecture. Besides, the above software and our work differ in scope. We focus more on providing MSA-based service composition in terms of IoT-Cloud service. However, looking at the direction of commercial software or other researches, we observe that our work that addresses both the cloud interoperability and the service composition is the right direction to move on. To summarize the relevant works mentioned above, we organize Table 1.

**Table 1.** Comparison of existing works on interoperability and service composition.

| Research Work | Interoperability | Multiple Clouds | MSA-Based Service Composition | Open-Source |
| --- | --- | --- | --- | --- |
| [17] | Yes | Yes | No | No |
| [18] | Yes | Yes | No | No |
| [19] | Yes | Yes | No | Yes |
| [20] | Yes | Yes | No | Yes |
| [21] | Yes | Yes | No | No |
| [22] | Yes | No | No | Yes |
| [23] | Yes | No | Partially yes | Yes |
| [24] | Yes | Yes | Partially yes | No |
| [25] | Yes | Yes | Partially yes | Yes |
| [26] | Yes | Yes | Yes | Partially yes |
| **Dynamic OverCloud** | **Yes** | **Yes** | **Yes** | **Yes** |

## 3. Dynamic OverCloud: Design

In this section, we describe an overall design of Dynamic OverCloud based on requirements for realizing IoT-Cloud service composition over multiple clouds. Also, we design a software framework to build the proposed concept dynamically.

### 3.1. Requirements

To satisfy the inter-operable and visibility supported specialized overlay layer that provides IoT-Cloud service composition with underlying clouds, the following requirements are discussed, as depicted in Figure 1.

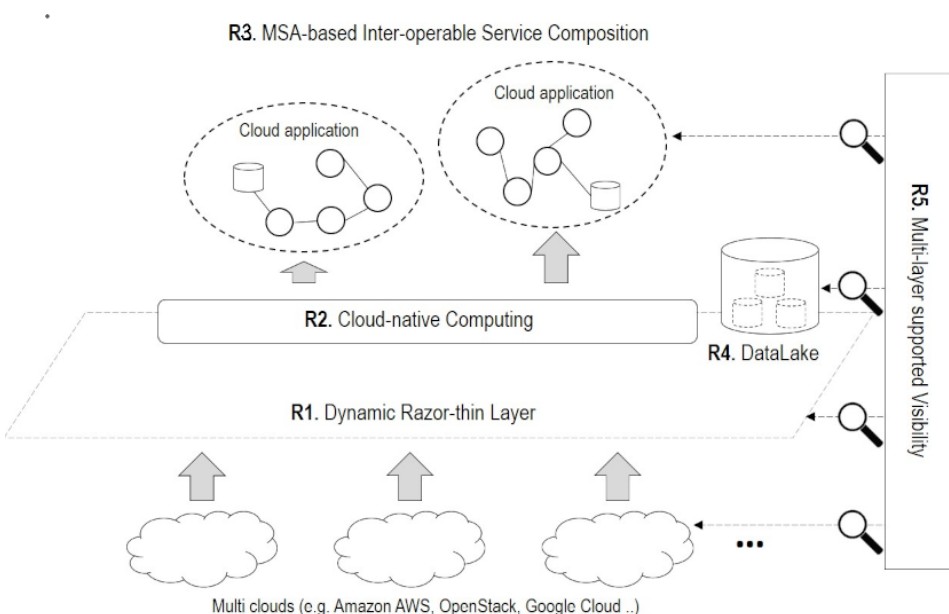

**Figure 1.** Requirements to solve the problem.

- **R1. Dynamic Razor-thin Layer:** Cloud interoperability is a critical mission to provide developers with flexible and powerful resources by avoiding vendor lock-in problem. The IoT-Cloud services are recommended to dynamically utilize near-by resources among multi-clouds for low-latency and high-throughput connectivity. Building an additional razor-thin layer that can be dynamically configured over multiple clouds is a simple way to meet requirements that satisfy these characteristics of the IoT-Cloud service. In particular, the additional layer should be lightweight and require less overhead. It also should work smoothly on any clouds.
- **R2. Cloud-native Computing:** Developers want to quickly and easily develop and validate cloud applications regardless of the underlying infrastructure. Cloud-native computing has changed

the way we deploy our services into the infrastructure since containerization enables developers to make lightweight isolation that can easily and quickly deploy their codes to realize services. Cloud-native computing mainly provides computing/networking/storage resources over the underlying infrastructure on top of Kubernetes orchestration. With the features of Container Networking Interface (CNI) and Container Storage Interface (CSI), containers that have stateless properties can be flexibly connected with the persistent storage in a standard way of cloud-native computing defined by CNCF(Cloud Native Computing Foundation).

- **R3. MSA-based Inter-operable Service Composition:** For performing IoT-Cloud services on cloud-native computing, We need service composition in the form of microservices architectures. Thus, service composition should remove the risk of any friction or conflict between dependency problems by taking advantage of containers. Besides, considering the mobility and geographical characteristics of IoT-Cloud services, scalability and fault-tolerance should also be considered for inter-operable service compositions.

- **R4. DataLake:** Collecting and Managing data is the most valuable thing for developing and validating services. Many kinds of data, such as service domain data (e.g., machine temperature, the humidity of a room, and so on), are generated by service composition. Also, operation data (e.g., resource utilization, path, links, logs and, etc.) to understand the situation of both services and infrastructures are generated. Raw data is also needed to perform specific functionalities. These data should be systemically stored and managed to get new insights at any time. In other words, we need integrated storage that leverages existing storage or configures new storage to keep their valuable data.

- **R5. Multi-layer supported Visibility:** Multi-layer visibility across resource, flow, workload layers is required for both developers and operators. From an operator's point of view, visibility measurement, collection, and associated visualization are essential for the continued operation of their infrastructure, so that operators can gain timely insights into the operational status of resources and associated flows [27]. The developer's point of view is that they also need a visibility solution to understand the situation of their services and enable better workload placement and optimized resource utilization for better service compositions.

*3.2. Overall Design of Dynamic OverCloud*

Based on the above requirements, we propose a new cloud-leveraged software concept called Dynamic OverCloud, as depicted in Figure 2. Dynamic OverCloud is an additional layer located between services and resource layers to provide a consistent environment in a cloud-agnostic way.

To satisfy the R1, we carefully design the Dynamic OverCloud overlay layer concept. The key feature of the Dynamic OverCloud layer is to ensure cloud interoperability at given underlay clouds. Leveraging cloud resources that are not dependent on particular cloud infrastructure is a challenging issue due to the characteristics of clouds [28]. Most of PaaS and SaaS services provided by cloud vendors are primarily dependent on specific providers. Therefore, instead of the whole effort for cloud interoperability, we leverage the minimal IaaS-level APIs for allocating cloud resources from cloud vendors. Also, for the R1, containerization is used to facilitates lightweight provisioning of Dynamic OverCloud with less overhead on any Linux-based cloud environment. Besides, we design a workflow-driven software to automatically perform overall tasks on the provisioning of Dynamic OverCloud in an efficient manner.

To handle the R2, we design Cloud-native Clusters as a component of Dynamic OverCloud. They are a collection of logical resources capable of cloud-native computing. They provide users with computing/networking/storage resources to running their services. Allocated resources are in the form of containers to facilitate microservices architecture.

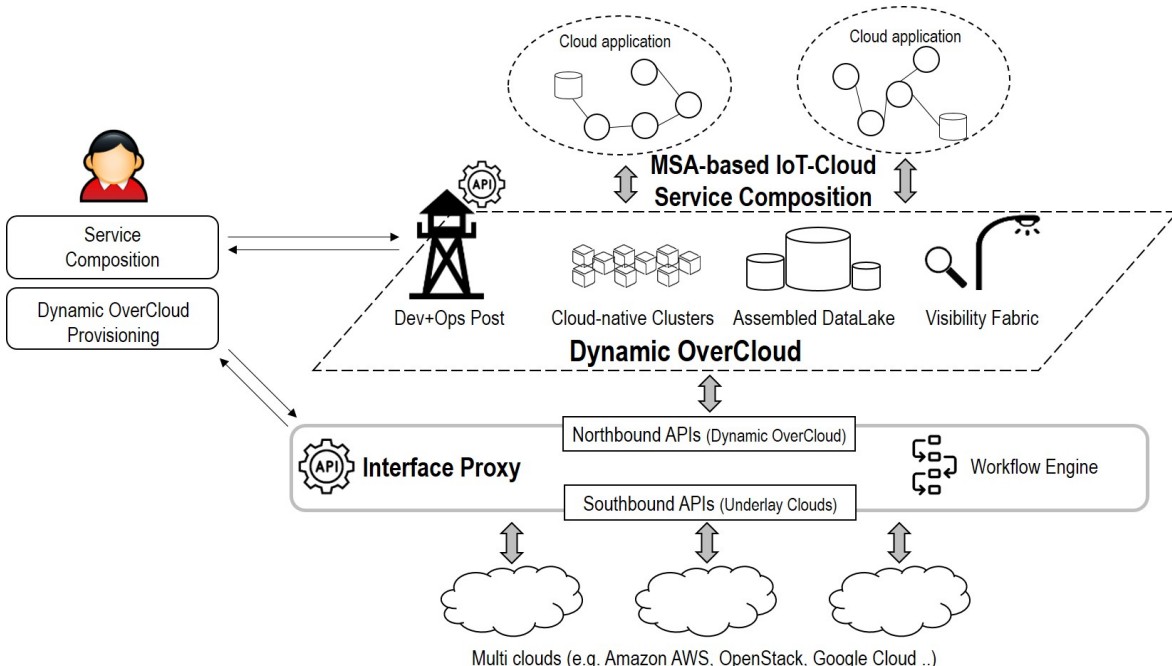

**Figure 2.** Overall concept of Dynamic OverCloud.

To cover the R3, we design Dev+Ops Post in Dynamic OverCloud, which enables inter-operable service compositions by using Cloud-native Clusters. Dev+Ops Post also leverages other components to gain the storage and visibility needed for service composition.

For the R4, we define Assembled DataLake in Dynamic OverCloud. Assembled DataLake is a collection of an integrated datastore where data is flexible and connected in Dynamic OverCloud. It can store a variety of data depending on the purposes. It also supports persistent volume to complement the stateless characteristics of containers.

To satisfy the R5, we design Visibility Fabric, a collection of delicate visibility points that can facilitate the support of the multi-layer visibility in Dynamic OverCloud. It checks the status of the resources in Dynamic OverCloud and helps to understand the situation of services.

Figure 3 illustrates the components and their functionalities of Dynamic OverCloud. Dynamic OverCloud consists of Dev+Ops Post, Cloud-native Clusters, Assembled DataLake, and Visibility Fabric. For the communication between users and underlay clouds, Interface Proxy is designed. The components are described in detail below.

- **Interface Proxy:** Interface Proxy is a communication channel between users and underlay clouds. Northbound APIs interpret the user's requirement and generate information to forward them to Interface Core. Southbound APIs acquire cloud resources of underlay clouds based on the generated data. Note that Interface Proxy is not a dynamically generated entity each time, but it is a shared entity for multiple users. Dynamic OverCloud APIs run automatically basic operations of Dynamic OverCloud based on the Dynamic OverCloud workflows description. All of the transactions of Dynamic OverCloud APIs are stored in the Dynamic OverCloud Repository. For example, when a user deploys Dynamic OverCloud, the related data such as Dynamic OverCloud ID, specifications of Cloud-native Clusters, Assembled DataLake, and Visibility Fabric are stored to the Dynamic OverCloud repository.
- **Dev+Ops Post:** Dev+Ops Post is provisioning and orchestration entities in conjunction with other components. It automatically provisions other components following requirements. Orchestration tools in Dev+Ops Post support container orchestration with visibility and persistent storage capabilities in a cloud-native way. It also coordinates Cloud-native Clusters as primary resources

in Dynamic OverCloud. Based on the operational data in the Assembled DataLake, Operational Visualization gives users a visual representation to grasp the overall situation of their services.

- **Cloud-native Clusters:** Cloud-native clusters provide users with pre-prepared resources for the container-based service composition, unlike cloud resources offered by underlay clouds. It can be any containers using container runtime engines that are compatible with the standard of CNCF. It also leverages CNI and CSI to provide users with complete ICT resources (Computing, Networking, and Storage). In the user's point of view, Cloud-native Clusters are a logical pool of resources supporting containers supervised by container orchestration.

- **Assembled DataLake:** Assembled DataLake consists of time-series databases and distributed persistent storage for operation/service data. Time-series databases primarily store operational data that includes visibility and log data from Visibility Fabric to catch the status of Dynamic OverCloud. Service-domain data that is generated by running applications are stored in distributed persistent storage. Since, running application on Dynamic OverCloud is in the form of containers, distributed persistent storage should be compatible with containers. In the case of the existing storage that can be compatible with Kubernetes orchestration, it can be used as a persistent volume of containers.

- **Visibility Fabric:** Visibility Fabric provides resource, flow, and workload layer visibility to understand the overall situation of Dynamic OverCloud with the help of visibility solutions. For that, it injects visibility data collectors into Cloud-native Clusters in the form of a lightweight agent. Visibility Fabric supports dynamic resource-centric visibility rather than fixed topology due to the nature of Dynamic OverCloud. It also covers the view of workloads that understand the relations of functions for performed services, as well as the view of resources and flow that checks the status of resources and networking in Dynamic OverCloud.

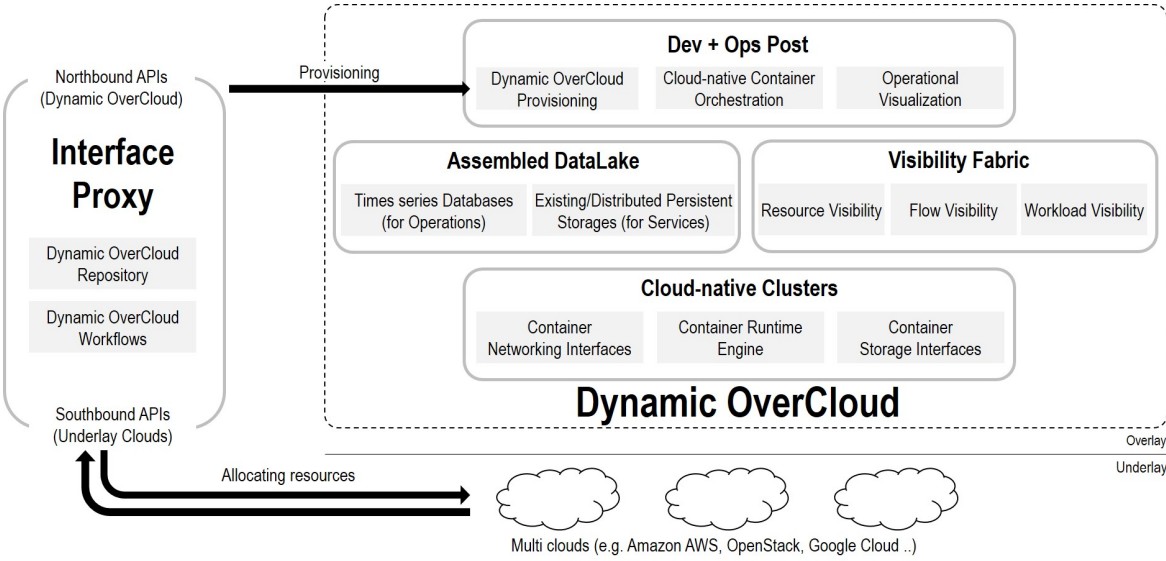

**Figure 3.** Dynamic OverCloud components design.

### 3.3. Software Framework Design of Dynamic OverCloud

Based on overall designed Dynamic OverCloud, we design a software framework that enables Dynamic OverCloud provisioning, as shown in Figure 4. The software framework mainly uses Interface Proxy and Dev+Ops Post components. To handle the software framework, we carefully design Interface Proxy Core additionally. Northbound APIs are a communication channel between users and the software framework. The requests that come from Northbound APIs go to the Interface Proxy Core. It internally communicates with Southbound APIs to acquire cloud resource authority from UnderCloud. Southbound APIs understand pre-defined plugins that wrap IaaS-level APIs

provided by cloud vendors. For the support of multiple users, we leverage ID authentication tools. Final provisioning requests are forward to Dev+Ops Post with workflow description. After the API requests, the software framework manages all of the transactions by storing them to Dynamic OverCloud Repository.

Dev+Ops Post automatically builds and coordinates other components of Dynamic OverCloud, depending on the associated workflows. Dev+Ops Post uses containerization to make components of Dynamic OverCloud as light as possible and to build a lightweight environment that is not affected by a specific UnderCloud. Once Cloud-native Clusters are ready by Dev+Ops Post, distributed storage that can be connected to Cloud-native Clusters with the help of CSI are prepared. Visibility Fabric injects visibility collectors to Cloud-native Clusters. After that, it continually sends operation visibility data to Assembled DataLake in the format of time-series data.

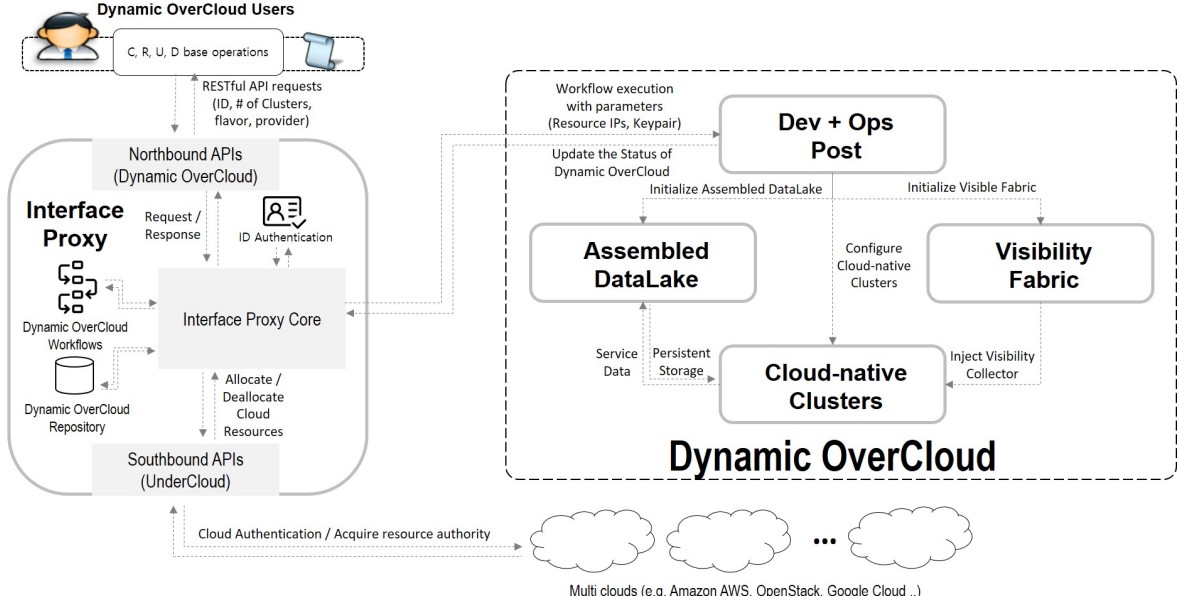

**Figure 4.** Design of software framework for building Dynamic OverCloud.

The workflow engine manages all operations of the software framework by describing workflows. The workflows can be performed automatically and in parallel according to the description. Thus, all of the procedures can have the advantage of automation and efficient handling in the software framework.

## 4. Dynamic OverCloud: Implementation

In this section, we provide an implementation of the proposed Dynamic OverCloud. First, we describe the components' implementation for the software framework. Then, we implement Dynamic OverCloud Workflow so that the designed software framework works for automated and efficient provisioning. The implementation can be found on Github (https://github.com/K-OverCloud/Dynamic-OverCloud) for everyone's use.

### 4.1. Software Framework Components' Implementation

Before we implement the designed software framework, we develop the software framework components first. Figure 5a shows the detailed Dynamic OverCloud components implementation. In Dev+Ops Post, we leverage Kubernetes for preparing cloud-native computing. Kubernetes can be run on any Linux-based environment [29]. However, it has mainly difficulty managing the Kubernetes cluster based on hybrid clouds since the main IPs in VMs are usually in a private network. For simplicity, rather than implementing the multi-clouds federation, we specially configure the advertising IP that is recognized as the external IP to build Kubernetes cluster over hybrid clouds.

To observe the overall status of Dynamic OverCloud, we leverage open-source-based visualization tools such as Chronograf, Grafana, and Prometheus for operation visualization [30,31].

For the Cloud-native Clusters, several container runtime engines are available. Among them, we choose the Docker since it mostly covers diversified applications range from 3-tier to IoT-Cloud, ML/DL, and partially HPC. CNI and CSI are used as assistance in conjunction with the Kubernetes to provide networking and storage resources. Since Cloud-native Clusters are mainly small and simple, we choose the Weave Net plugin with CNI for networking. For the storage resources in Cloud-native Clusters, we leverage Rook [32], a storage orchestrator in cloud-native computing. It internally uses CNI to adopt storage plugins.

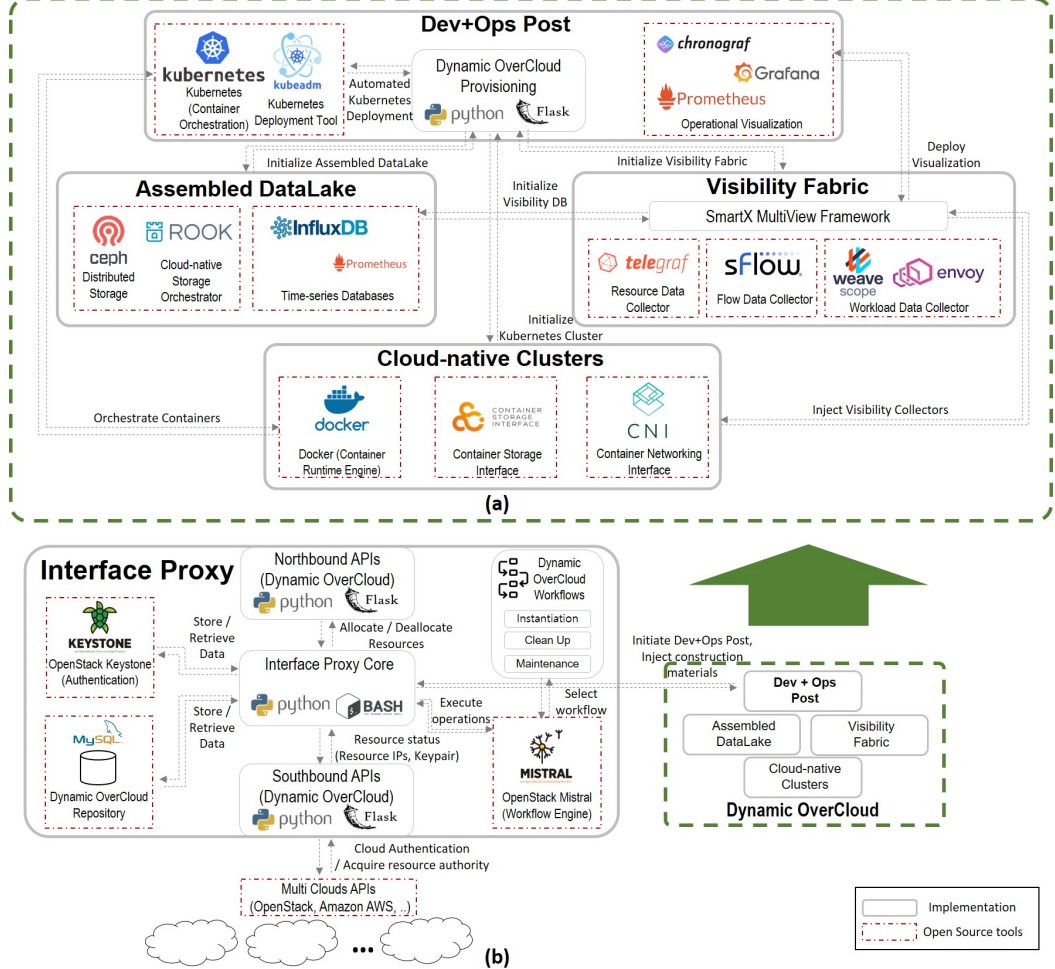

**Figure 5.** Components Implementation: (**a**) Dynamic OverCloud, (**b**) Interface Proxy.

For the Assembled DataLake, we deploy persistent storage with Rook. We use Ceph for storage provider options in Rook since it is highly stable and supports various solutions for block storage, object storage, and shared filesystems. Instead of using Rook to provision persistent storage dynamically, it is also possible to configure existing storage as a persistent volume in Cloud-native Clusters. Various storage plugins on Kubernetes can be found in [33]. After preparing the persistent storage, we deploy time-series databases by leveraging InfluxDB and Prometheus for operational data. Since time-series databases in Cloud-native Clusters make use of persistent storage, the order of implementation should be carefully considered.

For the Visibility Fabric, we apply the developed visibility software called as SmartX MultiView Framework in previous our work [34]. SmartX MultiView Framework has multiple stages such as Visibility Collection, Validation, Staging, Visualization to get operational status. We leverage Telegraf,

Sflow, Weave Scope, and Envoy as a visibility collector to get metrics and events from Cloud-native Cluster with the help of the SmartX MultiView Framework [35,36]. Visibility collectors are injected to Cloud-native Clusters in the form of Kubernetes pods.

Figure 5b shows the implementation of Interface Proxy. We use Python and Flask to implement the Northbound/Southbound APIs as the RESTful API for Dynamic OverCloud users. For the Interface Proxy Core, we implement its functionalities using bash and python scripts. For the Dynamic OverCloud Repository, we use MySQL to store the transactions. To implement the automated procedure of basic operations, OpenStack Mistral is used to a workflow engine. It is independent of the OpenStack environment so that it can perform a workflow service without specific cloud constraints. We use a reverse workflow scheme in the form of DSL (Domain Specific Language) provided by OpenStack Mistral. The reverse workflow can depend on each task so that they can work in parallel while synchronizing between tasks. DSL is a YAML Ain't Markup Language (YAML) format. The workflow engine performs the specified tasks according to the description of DSL. For each task in the workflow, we implement a bash/python script with automated provisioning tools. Interface Proxy Core injects implemented bash/python scripts that can be executed through SSH into Dev+Ops Post by complying with the selected workflow. Detailed workflows are covered in the following subsection.

### 4.2. Dynamic OverCloud Workflows Implementation with the Software Framework

The developed components are performed by defined workflows to complete with the software framework. As shown in Figure 6, we implement the Dynamic OverCloud Workflows for basic operations such as instantiation, clean-up, and maintenance. The blue rectangles in Figure 6 represent tasks in workflows that are implemented with python/bash scripts. These tasks are executed as a unit of work in the workflow. That is, the collection of the tasks between the Interface Proxy Core are actual workflows in the form of DSL. The implemented workflows are executed in the internal process of the RESTful API, so the user does not need to know the details of the workflows. Note that we only deal with OpenStack, Amazon AWS, and Hybrid cloud(OpenStack and Amazon AWS) for the implementation.

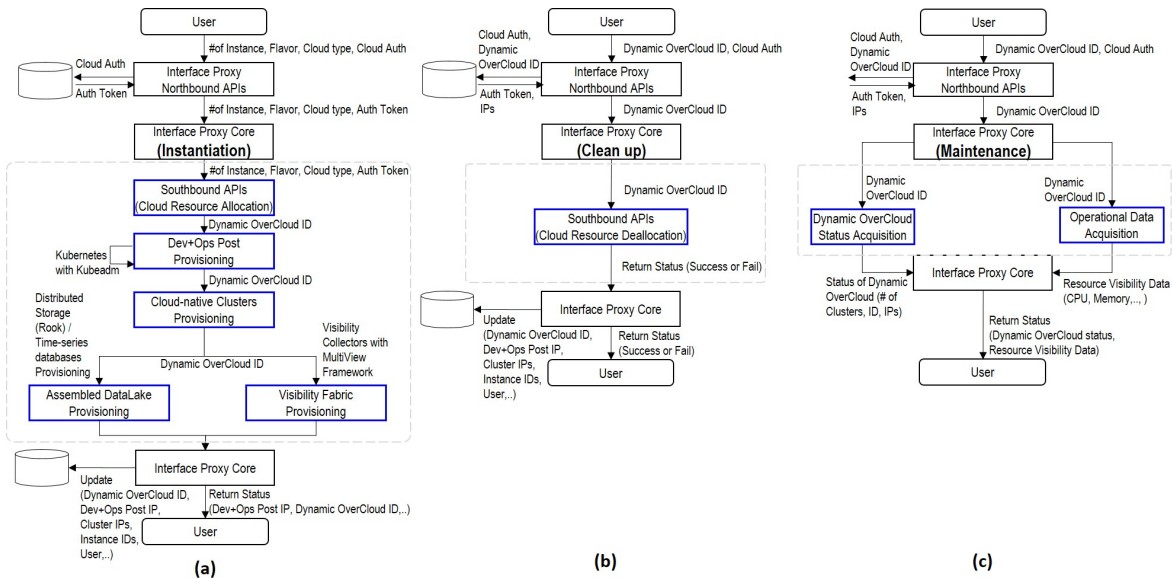

**Figure 6.** Dynamic OverCloud Workflows: (**a**) Instantiation, (**b**) Clean up, (**c**) Maintenance.

For Figure 6a, we carefully implement several tasks. In the cloud resource allocation tasks, we implement cloud resource allocation by leveraging cloud provider's IaaS-level APIs. Once the cloud authentication is confirmed, it allocates cloud resources depending on cloud types. After allocating cloud resources, it creates transaction data such as Dynamic OverCloud ID, allocated resource IPs.

The Dynamic OverCloud ID is used as an input parameter for the following tasks. In Dev+Ops Post and Cloud-native Clusters provisioning tasks, we implement automated Kubernetes deployment with the Kubeadm tool. The implemented script makes use of the Dynamic OverCloud IDs to find allocated IPs and SSH key data from the Dynamic OverCloud Repository. For Assembled DataLake provisioning, we implement automated deployment for persistent storage by leveraging Rook. We also develop time-series database provisioning after the persistent storage provisioning. In Visibility Fabric provisioning task, we deploy visibility collectors with the SmartX MultiView framework on the Cloud-native Clusters. When all tasks are completed, the final transaction is stored in the Dynamic OverCloud Repository and the results are forwarded to the user. Since Assembled DataLake and Visibility Fabric tasks proceed in parallel with the help of reverse workflow, automated and efficient provisioning is possible.

Figure 6b,c are quite simple rather than the Figure 6a. For the Cloud Resource Deallocation task in Figure 6b, we leverage cloud provider's APIs to deallocate cloud resources. Likewise, the Dynamic OverCloud ID is used to find the information related to allocated resources. After deallocating cloud resources, transactions are updated. For Figure 6c, two tasks are developed. In the Dynamic OverCloud Status Acquisition task, we implement the script to find and return the transaction information related to the given Dynamic OverCloud ID. For the Operation Data Acquisition task, we develop the script to retrieve resource visibility data from time-series databases in Assembled DataLake based on the Dynamic OverCloud ID. These two tasks are also executed simultaneously.

## 5. Dynamic OverCloud: Feasibility Verification with Operation Lifecycle

In this section, we introduce an operation lifecycle using the proposed Dynamic OverCloud for realizing IoT-Cloud service composition. Then, we verify service composition with the operation lifecycle by selecting a real-world IoT-Cloud service scenario. After that, we discuss the feasibility of Dynamic OverCloud.

### 5.1. Operation Lifecycle on Dynamic OverCloud

To guide the process from provisioning Dynamic OverCloud to service realization, we suggest an operation lifecycle, as depicted in Figure 7. The provisioning stage is done automatically with the help of the implemented software framework, but Dynamic OverCloud users must manually perform the stages of service composition and service verification. Thus, the Dynamic OverCloud user should comply with following Operation Lifecycle to utilize Dynamic OverCloud for service composition effectively.

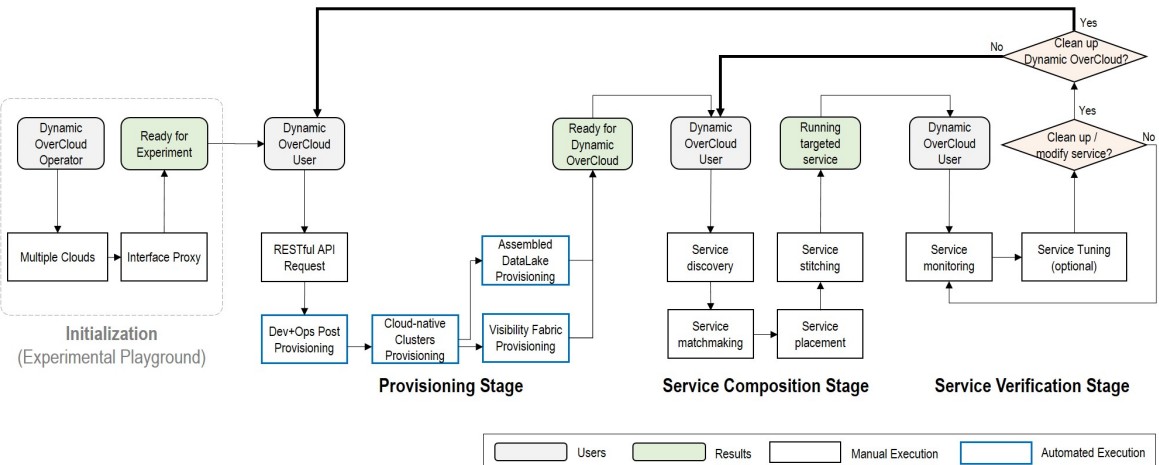

**Figure 7.** Operation Lifecycle for service realization.

Before starting the operation lifecycle, we should prepare the experimental environment. In this step, users should acquire cloud credentials to create bare metal/virtual machines with an isolated tenant network. For the ready to use of Interface Proxy, we install and configure a database and a workflow engine. Also, we configure the credentials of targeted clouds to use of resources.

Provisioning Stage: This stage is to build Dynamic OverCloud for the ready to use. It automatically provisions Dynamic OverCloud step by step with the workflow engine based on the user's requirements.

The user should call RESTful API request with parameters as depicted in Table 2. When the API request comes in, Interface Proxy executes the appropriate workflow depending on the request parameters to build Dynamic OverCloud. After building Dynamic OverCloud, the user receives a response result, as shown in Figure 8. By accessing the Dev+Ops Post with the given SSH key, the user is ready to use Dynamic OverCloud. Multiple dashboards that are given by the API response are also available to assist the users in utilization.

**Table 2.** Provisioning API specification.

| Parameter | Description |
| --- | --- |
| Provider | Cloud type (OpenStack, Amazon, heterogeneous) |
| Number | The number of cloud-native clusters |
| Size | Instance flavor |
| OpenStack.number | The number of cloud-native clusters (heterogeneous) |
| OpenStack.size | Flavor (heterogeneous) |
| OpenStack.post | Dev+Ops Post location (yes, no) |
| Amazon.number | The number of cloud-native clusters (heterogeneous) |
| Amazon.size | Flavor (heterogeneous) |
| Amazon.post | Dev+Ops Post location (yes, no) |

```
{
    "overcloud_ID": "10fd3e7c-5298-4880-9904-480aace97302",
    "weave_url": "http://$Dev+Ops Post IP:32080",
    "devops_post": "$Dev+Ops Post IP",
    "chronograf_url": "http://$Dev+Ops Post IP:8888",
    "ssh": "-----BEGIN RSA PRIVATE KEY-----...-----END RSA PRIVATE KEY---",
    "prometheus_url": "http://$Dev+Ops Post IP:30921",
    "rook_url":  "http:// $Dev+Ops Post IP:32524",
    "logical_cluster": [ " $Cluster IP #1",
                         " $Cluster IP #2",
                         …],
    "smartx-multiview": "http://$Dev+Ops Post IP:3006/menu"
}
```

**Figure 8.** Example response of Provisioning API.

Service Composition Stage: This stage is to do service composition to validate user's services. Since the service composition is limited to Cloud-native Clusters, we utilize the steps of service composition on the Kubernetes environment defined in our previous work [37]. For IoT-Cloud service composition, the following four steps are considered.

First, the service discovery step should be considered for a developed service. A service based on MSA is essential to recognize and communicate with each other. In Kubernetes, we find functions by using the metadata in the service description. To discover functions from outside, we use the service object in Kubernetes. It chooses multiple options such as node port and the ingress IP to expose functions. Thus, we should carefully write the service description by using these functionalities to satisfy service discovery.

Second, finding and matching the resources is important to accord with the most suitable requirements of the service function among the various resources. Each function for composition has a different resource requirement and the location of the target resource to be executed. In this step, we find the most appropriate resources to distribute each of the functions of the desired service. We select the factors and apply them by writing in the service description.

Third, after service matchmaking, service placement should be considered. If there is a required precedence function in the service, it should be arranged in order. The service distribution should be done after checking the node status of the previously matched functions. At this time, the relation of the separated functions must be specified in order to prevent a crash between the functions. The service distinguishes functions through attached labels and distributes the functions by separating them through the namespace in the description of the service to be deployed.

Lastly, the distribution functions are linked according to the service specification. This linkage enables us to transmit and receive data of mutually connected functions by defining the connection relationship of functions. When the dependency of the service function is satisfied, we define the relationship between the functions by specifying the label and the selector in the Kubernetes description.

Service Verification Stage: In this stage, we check whether the service is performing normally or not, and have troubleshooting if a problem occurs. After the service stitching, the targeted service is operated. However, continuous monitoring and service management are needed. Service monitoring in a cloud-native computing environment should be at the level of modularized service functions. The service is maintained by continuously checking the status of service functions and taking action accordingly. In addition, maintaining continuous service is necessary by taking action if the status of the service functions is abnormal. Service tuning in the cloud-native computing environment improves qualities by re-distributing and re-stitching modularized functions.

### 5.2. IoT-Cloud Service Realization with Operation Lifecycle

To verify the proposed Dynamic OverCloud, we select smart energy IoT-Cloud service as a real-world service scenario since it covers range from 3-tier and IoT to machine learning. Smart energy service uses Raspberry Pi 2 to collect temperature, humidity, and power consumption of the server room and collects the server's system temperature. If the service detects an abnormality such as a high temperature or excessive power consumption of the server, it changes servers to a power-saving mode or notifies a server administrator. It also has visualization and monitoring on a dashboard via a web browser [38]. We make use of the designed Lifecycle workflow of Dynamic OverCloud to realize the IoT-Cloud service as follows.

Before starting the operation lifecycle, we prepare an experimental playground, as shown in Figure 9. In this step, users should acquire cloud credentials to create bare metal/virtual machines with an isolated tenant network. For the ready to use of Interface Proxy, we install and configure a database and a workflow engine. Also, we configure the credentials of targeted clouds to use of resources.

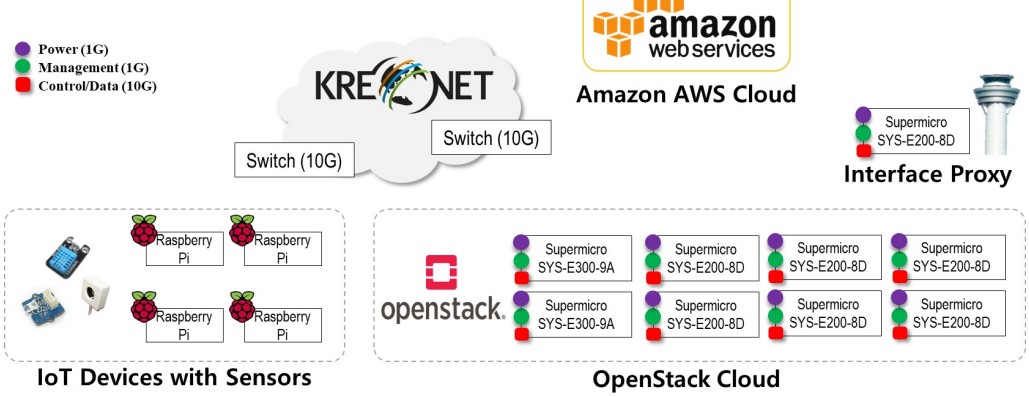

**Figure 9.** Experimental playground for IoT-Cloud service over multiple clouds.

To verify Dynamic OverCloud with the implemented software framework, we prepare the multi-clouds playground shown as Figure 9. Using multiple clouds, we use Amazon AWS for the public cloud and build our private OpenStack-based cloud. We use six SuperServer E200-8D (Xeon-D 1.9Ghz, 6-cores, 32GB memory, 512G SSD) and two SuperServer E300-9A (Xeon-D 2.2Ghz, 4-cores, 32GB memory, 512G SSD) boxes to build the OpenStack cloud and all of servers are manufactured by SuperMicro, headquarters in the United States. One server with the SuperServer E200-8D model plays the role of Interface Proxy. In addition, we configure four Raspberry Pi 2, which is manufactured by Raspberry Pi Ltd from England, to realize the smart energy service. For the networking of the playground, we use three networking planes by categorizing heterogeneous traffics based on their characteristics; a power plane and a management plane for operators; a control/data plane for Dynamic OverCloud users.

For preparing Interface Proxy, we configure OpenStack Mistral engine and MySQL. Note that OpenStack Mistral can be run any environment without dependencies of the OpenStack cloud. Before running APIs, we should define parameters within the configuration file. The parameters we fill out are listed in Table 3.

**Table 3.** Interface Proxy configuration parameters.

| | Initial Parameters |
|---|---|
| Database | MySQL host, password |
| OpenStack | OpenStack ID, password, Keystone url |
| Amazon AWS | Amazon AWS access key, secret access key |
| Workflow engine | OpenStack Mistral host, ID, password |

### 5.2.1. Provisioning Stage

In order to allocate cloud resources in Amazon AWS and OpenStack clouds built in the previous stage, we call implemented Dynamic OverCloud API with the following command.

```
$ curl –X POST –H "Content-Type: application/json" –d '{"provider": "heterogeneous", "OpenStack": {"number": "2", "size": "m1.logical", "post": "no"}, "Amazon": {"number": "3", "size": "c5d.2xlarge", "post": yes"}}' http://Interface_Proxy_IP:6125/overclouds
```

This command is the simplest way to build Dynamic OverCloud at once. It means that we build Dynamic OverCloud with two virtual machine instances with m1.logical size on OpenStack cloud and three virtual machines with c5d.2xlarge size on Amazon AWS cloud. Dev+Ops Post will be deployed on Amazon AWS, and size is fixed to cover significant computing power. When the API request comes in, Interface Proxy selects the appropriate workflow according to the user's input request and performs it automatically. At that time, UnderCloud allocates one additional virtual machine for Dev+Ops Post. After the provisioning of resources, scripts are executed automatically in virtual machines to construct the components of Dynamic OverCloud. Dev+Ops Post executes implemented scripts by synchronizing the task order according to the workflow description. After the provisioning of the Dynamic OverCloud, API returns the following information.

```
$ { "devops_post": "IP", "ssh": "PRIVATE_KEY", "logical_cluster": [ "IP1", "IP2", "IP3", "IP4", "IP5"], "overcloud_ID": "ID", "Prometheus_url": "URL", "rook_url": "URL", "chronograf_url": "URL", "weave_url": "URL", "smartx-multiview": "URL"}
```

After that, we create an ssh-key file by pasting the ssh tuple's content to access Dev+Ops Post and Cloud-native Clusters using the SSH. In the Dev+Ops Post, we see the Kubernetes and pods in Dev+Ops Post, as shown in Figure 10. We confirm five Cloud-native Clusters and several pods to assist the functionalities of Dynamic OverCloud.

To observe the overall resource status of Dynamic OverCloud, we use web visualization URLs. By using the tuple of chronograf_url of the output of the API call, we access resource-layer visualization.

It is implemented through data collection of Telegraf, a database of InfluxDB, and visualization of Chronograf. The collected metrics are basic resources status such as CPU, disk, diskio, kernel, memory, process, swap, system.

For checking persistent storage, we access the Ceph web dashboard. Ceph dashboard shows five object storage daemons (OSDs) to make a total 605G storage pool for service data. Rook orchestrates back-end storage drivers such as Ceph in a cloud-native environment. Note that Rook basically uses the local storage on virtual machines to make the dynamic distributed storage pool. However, we can deploy Ceph storage with only dedicated nodes by using the label properties of Kubernetes.

```
ubuntu@ip-          :~$ kubectl get nodes -o wide
NAME               STATUS   ROLES    AGE    VERSION   INTERNAL-IP   EXTERNAL-IP   OS-IMAGE
ip-                Ready    <none>   3m25s  v1.12.1                 <none>        Ubuntu 16.04.5 LTS
ip-                Ready    <none>   3m34s  v1.12.1                 <none>        Ubuntu 16.04.5 LTS
ip-                Ready    master   3m56s  v1.12.1                 <none>        Ubuntu 16.04.5 LTS
ip-                Ready    <none>   3m22s  v1.12.1                 <none>        Ubuntu 16.04.5 LTS
logical-cluster-1  Ready    <none>   3m31s  v1.12.1                 <none>        Ubuntu 16.04.4 LTS
logical-cluster-2  Ready    <none>   3m26s  v1.12.1                 <none>        Ubuntu 16.04.4 LTS

ubuntu@ip-1             :~$ kubectl get pods --all-namespaces
NAMESPACE         NAME                                     READY   STATUS    RESTARTS   AGE
default           ambassador-85478b9f6d-qztpc              2/2     Running   0          2m59s
default           prometheus-operator-6c6cc6f56d-5dbcc     1/1     Running   0          3m6s
default           prometheus-prometheus-0                  2/2     Running   0          2m32s
kube-system       coredns-576cbf47c7-qlzhn                 1/1     Running   0          3m50s
kube-system       coredns-576cbf47c7-s54l6                 1/1     Running   0          3m50s
kube-system       etcd-ip-                                 1/1     Running   0          2m51s
kube-system       kube-apiserver-ip-                       1/1     Running   0          3m5s
kube-system       kube-controller-manager-ip-              1/1     Running   0          3m10s
kube-system       kube-proxy-dbtxx                         1/1     Running   0          3m39s
kube-system       kube-proxy-jwc6v                         1/1     Running   0          3m50s
kube-system       kube-proxy-kf75x                         1/1     Running   0          3m36s
kube-system       kube-proxy-kfvzv                         1/1     Running   0          3m40s
kube-system       kube-proxy-ltpdf                         1/1     Running   0          3m45s
kube-system       kube-proxy-wqjfc                         1/1     Running   0          3m48s
kube-system       kube-scheduler-                          1/1     Running   0          2m59s
kube-system       weave-net-5d6jp                          2/2     Running   0          3m32s
kube-system       weave-net-lppg4                          2/2     Running   0          3m32s
kube-system       weave-net-nbl2c                          2/2     Running   0          3m32s
kube-system       weave-net-x4pqx                          2/2     Running   0          3m32s
kube-system       weave-net-xmb5w                          2/2     Running   0          3m32s
kube-system       weave-net-zqdnh                          2/2     Running   0          3m32s
rook-ceph-system  rook-ceph-agent-2pmcs                    1/1     Running   0          2m20s
rook-ceph-system  rook-ceph-agent-blwnm                    1/1     Running   0          2m20s
rook-ceph-system  rook-ceph-agent-bqcng                    1/1     Running   0          2m20s
rook-ceph-system  rook-ceph-agent-m7pm9                    1/1     Running   0          2m20s
rook-ceph-system  rook-ceph-agent-qx4f2                    1/1     Running   0          2m20s
rook-ceph-system  rook-ceph-operator-7dd46f4549-86bkp      1/1     Running   0          2m40s
rook-ceph-system  rook-discover-6bn4c                      1/1     Running   0          2m20s
rook-ceph-system  rook-discover-bkgtx                      1/1     Running   0          2m20s
```

**Figure 10.** Status of Cloud-native Clusters in Dev+Ops Post of Dynamic OverCloud.

### 5.2.2. Service Composition Stage

In this stage, we leverage the developed smart energy IoT-cloud service to apply the service composition. First, the functions in the smart energy service are Python scripts developed with a microservice architecture shown as Figure 11. All of the functions are in the form of a Docker container. The detailed implementation is written in our previous work [37].

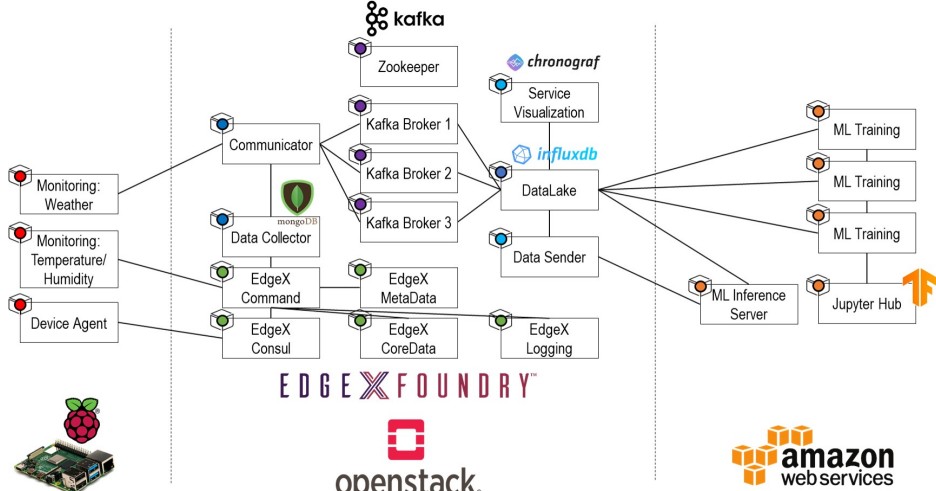

**Figure 11.** Implemented smart energy IoT-Cloud service function diagram.

For the service discovery step, the functions of smart energy service are divided into internal discovery and external discovery according to the access policy. The smart energy service has Kafka, DataLake, and Visualization functions that service discovery is done according to the internal access policy by describing metadata of service. In contrast, the Communicator, EdgeX service, Data collector, and ML functions, which are external discovery approaches, utilize NodePort on Kubernetes.

In the step of the service matchmaking, the Kubernetes deployment controller is used to make requests and limits on required resources by specifying them in the YAML file. For example, the Communicator function matches computing resources with 200 Milli-cores CPU and 64 Mega-byte memory requirements. Through this process, appropriate nodes can be matched with functions by considering needs.

In the service placement step, specific labels are assigned to Cloud-native clusters. We use the label to distribute service functions to the appropriate Cloud-native clusters. For example, the ML training functions that need high-performance computing are placed to Cloud-native Clusters, where are Amazon AWS cloud. We also proceed through the grouping of dependent functions. The grouping is done through the namespace, which is intended to prevent conflicts between the distributed functions.

Finally, in the service stitching step, the connections between functions are defined using the Kubernetes Selector in the YAML file. Also, we use the Kubernetes NodePort to connect with each other. For example, the inference function uses NodePort through service discovery to support inference requests based on RESTful API. It sends six consecutive datasets with five fields, such as external temperature, external humidity, server room temperature, server room humidity, and weather information, to the connected port in JSON format.

### 5.2.3. Service Verification Stage

In this stage, we should check the status of the container-based functions using the basic open-source visualization tool. In Dynamic OverCloud, many visualization tools are basically provided with the help of Visibility Fabric. Typically, workload-layer visibility focuses on the visualization of your container status and traffic connectivity in MSA-based service composition. Weave Scope is a representative tool to get the status of containerized functions. Figure 12 shows a graph-based visualization of the relationship between workloads depending on service stitching by accessing weave_url.

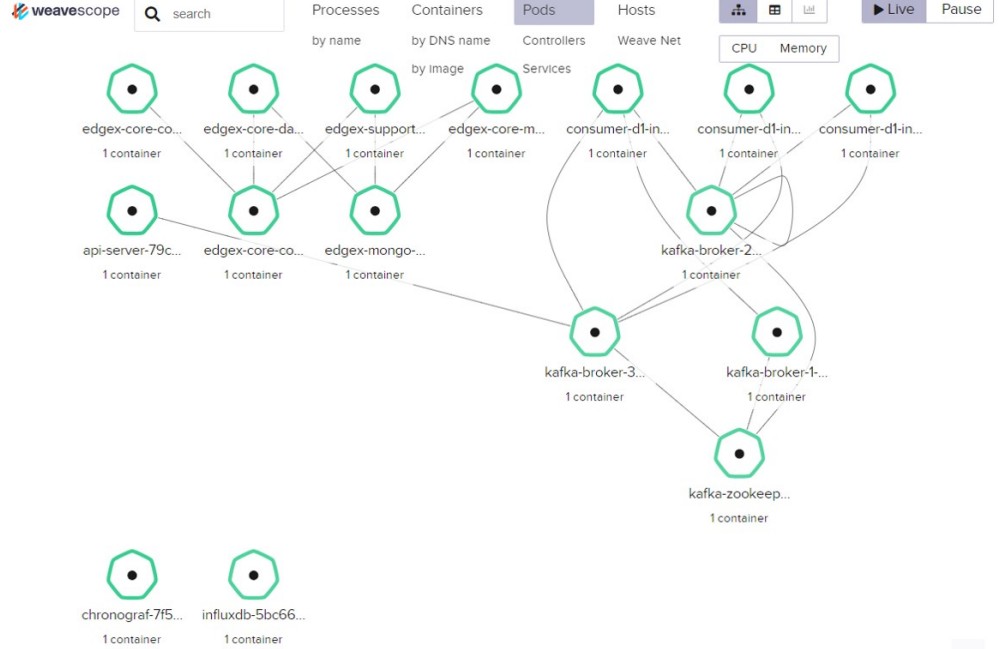

**Figure 12.** Workload-layer visualization in the smart energy service.

In another way, the service may have its service visibility. In our service case, we have web-based visualization to easily grasp the situation of the server room, as shown in Figure 13.

If there are problems among the inter-connected functions, we can seamlessly update functions as well as inter-connection by changing the Kubernetes Deployment description. Also, we can apply re-distribution policies when there are ongoing problems.

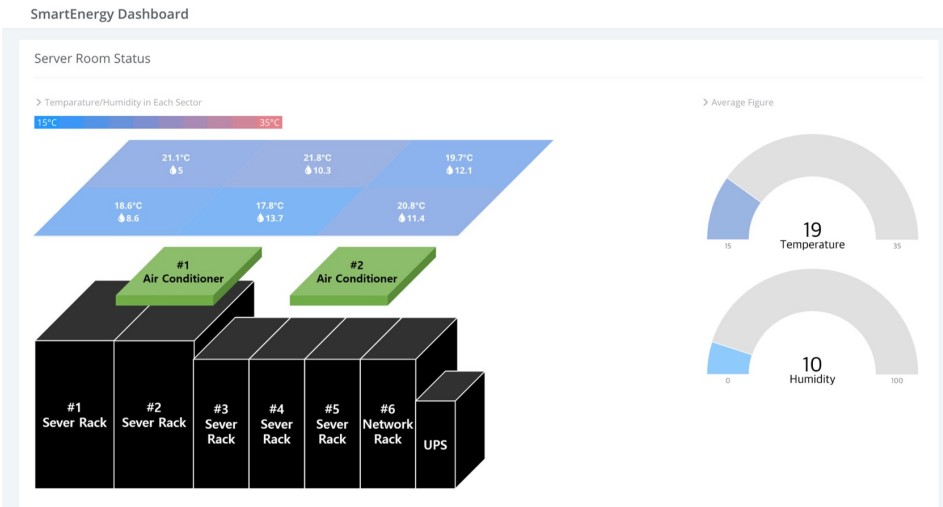

**Figure 13.** Visualization of the smart energy service.

*5.3. Discussion*

We verify the feasibility of the proposed Dynamic OverCloud by realizing the developed smart energy service. Thanks to the Kubernetes orchestration, the step of service composition can be done regardless of the underlying infrastructure. Once you build Dynamic OverCloud on hybrid clouds, you can easily place functions by considering the properties of the service. For example, in our smart energy service case, functions that should be adjacent to IoT devices are placed in our OpenStack cloud. In contrast, other functions that require a lot of computing are placed in the Amazon AWS cloud for efficient performance. Also, Kubernetes supports seamless updates using rolling updates, canary deployment. Based on these capabilities, users do the relocatable service composition on multi-clouds.

In the case of the vendor lock-in, Dynamic OverCloud currently supports OpenStack and Amazon AWS clouds. However, there is no need for much effort to support other clouds since we only wrap the minimal APIs for allocating/deallocating cloud resources from cloud providers. Also, All of the components of Dynamic OverCloud are fully open-source software. Thus, based on Kubernetes characteristics without vendor lock-in, we expect Dynamic OverCloud to satisfy more cloud interoperability by applying other clouds in the future.

To prove the efficient provisioning with the software framework, we run APIs to create and delete Dynamic OverCloud on the experimental playground. For performance comparison, we additionally execute instantiation and clean-up operations without workflow. Figure 14 shows the average times when instantiation and clean up are performed ten times. We verify that the instantiation operation with workflow could be done less than 7 minutes. In particular, provisioning time shows less overhead since all of the processes are in parallel by utilizing the workflow's characteristics.

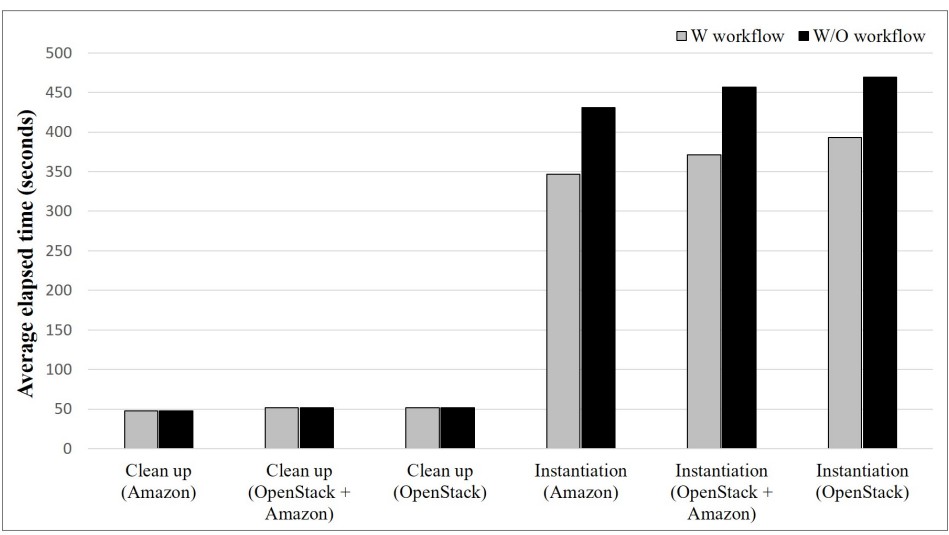

**Figure 14.** Time measurements of Dynamic OverCloud provisioning.

## 6. Conclusions

In this paper, we propose a new approach called Dynamic OverCloud that dynamically constructs inter-operable and visibility supported isolation to enable containerized MSA-based service composition. To verify the proposed concept, we design Dynamic OverCloud and software framework based on requirements. Then, we implement the proposed Dynamic OverCloud. To verify the feasibility of the proposed concept, we realize smart energy service with the operation lifecycle. Since we reflect automation and efficiency as much as possible into Dynamic OverCloud by using workflows, the user can dynamically deploy Dynamic OverCloud for their service verification. We also expect that anyone can easily apply Dynamic OverCloud to their environment without additional cost since Dynamic OverCloud is implemented by leveraging open-source software.

Although we verify the proposed concept with Smart energy service, this research has potential to be leveraged for various cloud applications across different domains such as SDN/NFV testbed for 4G/5G networks. In the future, we plan to expand and verify our software framework to apply the proposed concept in 5G mobile network environment.

**Author Contributions:** Conceptualization, J.H., S.P. and J.K.; Investigation, J.H.; Software, J.H.; Supervision, J.K.; Validation, J.H.; Writing—original draft, J.H.;Writing—review & editing, J.H., S.P. and J.K. All authors have read and agreed to the published version of the manuscript.

**Funding:** This work was supported by GIST Research Institute (GRI) grant funded by the GIST in 2020 and Institute of Information & communications Technology Planning & Evaluation (IITP) grant funded by the Korea government (MSIT) (No.2019-0-01842, Artificial Intelligence Graduate School Program (GIST)).

**Conflicts of Interest:** The authors declare no conflict of interest.

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
