# Peer review of "Dynamic OverCloud: Realizing Microservices-Based IoT-Cloud Service Composition over Multiple Clouds"

_electronics, doi:10.3390/electronics9060969_

Round 1

Reviewer 1 Report

The proposed architecture and its implementation is interesting, original and worth to be reported. 

Several improvements to the presentation are suggested:

(1) sections cannot start with a figure (move it after the citation)

(2) table 1 is not relevant and can be replaced with the citation

(3) The message of figures like 10 to 12 (or the bottom of 9) are not clear and not necessary

(4) The positioning vs. the existing approaches is very weak, especially in what concerns the experiment.

Reviewer 2 Report

The authors address the vendor lock-in problems that limit the inter-operable service compositions across multiple cloud offerings. They propose an overlay approach to address the vendor lock-in problem, which provides cloud users with an inter-operable and visibility-supported environment for micro-services architecture-based IoT-Cloud service composition over the existing multiple clouds. The authors present several conceptual frameworks that dynamically builds the proposed concept. Part of the detailed conceptual frameworks was prototyped and smart energy IoT-Cloud service used to verify feasibility. Overall, the authors make useful contribution. The introduction should include motivation. My concerns is that the issues dealt with have multiple dimensions and I believe that the authors do not really go in detail in each area. The paper is based on paper that are a bit old. It would be nice if the authors base the problem statement on newer work showing that there still gaps. Also, in related work, rather than just listing what the other did, it would be nice if you could critically analyse them. I know that you did present it in the last paragraph somewhat but would like to see for each one. Please include recent references as well.

Reviewer 3 Report

The authors propose a novel overlay approach Dynamic OverCloud which is an overlay layer that provides users with an inter-operable and visibility-supported environment for IoT-Cloud service composition over multiple clouds.

The authors connect their contribution to the concept of Infrastructure as a Service (IaaS). This can be further extended to 5G networks and service orchestration across different domains [1]. The authors are encouraged to add a paragraph in the manuscript where they explain the potentials of their framework for next generation mobile networks. 

[1] "A Cloud-based SDN/NFV Testbed for End-to-End Network Slicing in 4G/5G" by A. Esmaeily, K. Kralevska, D. Gligoroski

Another suggestion for future work is applying the framework to satisfy the requirements for other 5G relevant use cases.

Proof reading is needed. For instance, please correct "Cloud-native computing mainly provides computingnetworkingstorage resource... " (p. 4).

The authors can provide a link to GitHub repository for the proposed framework. It is correct that it is build from open-source software, but the impact of this work will be even bigger if it is made open-source.

The visibility of some of the figures, such as Fig. 11 and 14, has to be improved.

As a summary, it is a significant contribution to the state-of-the-art and the authors should make visible the potentials of their work.
